

# Estimation of the high-resolution variability of extreme wind speeds for a better management of wind damage risks to forest-based bioeconomy

Ari K. Venäläinen[1], Mikko O. Laapas[1], Pentti I. Pirinen[1], Matti Horttanainen[1], Reijo Hyvönen[1], Ilari Lehtonen[1], Päivi Junila[1], Meiting Hou[2], Heli M. Peltola[3]

[1] Climate Service Centre, Finnish Meteorological Institute, Helsinki, FI-00101, Finland
[2] China Meteorological Administration Training Centre, Beijing 100081, China
[3] School of Forest Sciences, University of Eastern Finland, Joensuu, FI-80101, Finland

*Correspondence to*: Ari K. Venäläinen (ari.venalainen@fmi.fi)

**Abstract** The bioeconomy has an increasing role to play in climate change mitigation and the sustainable development of national economies. In a forested country, such as Finland, over 50% of its current bioeconomy relies on the sustainable management and utilization of forest resources. Wind storms are a major risk that forests are exposed to and high spatial resolution analysis of the most vulnerable locations can produce risk assessment of forest management planning. Coarse spatial resolution estimates of the return levels of maximum wind speed based, e.g., on reanalysed meteorological data or climate scenarios can be downscaled to forest stand levels with the help of land cover and terrain elevation data. In this paper, we examine the feasibility of the wind multiplier approach for downscaling of maximum wind speed, using 20 meter spatial resolution CORINE-land use dataset and high resolution digital elevation data. A coarse spatial resolution estimate of the 10-year return level of maximum wind speed was obtained from the ERA-Interim reanalysed data. These data were downscaled to 26 meteorological station locations to represent very diverse environments: Open Baltic Sea islands, agricultural land, forested areas, and Northern Finland treeless fells. Applying a comparison, the downscaled 10-year return levels explained 77% of the observed variation among the stations examined. In addition, the spatial variation of wind multiplier downscaled 10-year return level wind was compared with the WAsP- model simulated wind. The heterogeneous test area was situated in Northern Finland, and it was found that the major features of the spatial variation were similar, but in the details, there were relatively large differences. However, for areas representing a typical Finnish forested landscape with no major topographic variation, both of the methods produced very similar results. Further fine-tuning of wind multipliers could improve the downscaling for the locations with large topographic variation. However, the current results already indicate that the wind multiplier method offers a pragmatic and computationally feasible tool for identifying at a high spatial resolution those locations having the highest forest wind damage risks. It can also be used to provide the necessary wind climate information for wind damage risk model calculations, thus making it possible to estimate the probability of predicted threshold wind speeds for wind damage and consequently the probability (and amount) of wind damage for certain forest stand configurations.



## 1 Introduction

The forest-based bioeconomy plays an important role in climate change mitigation (Kilpeläinen et al., 2016), and in a forested country like Finland, over 50% of the current bioeconomy there relies on the sustainable management and utilization of forest resources. A warming climate is expected to increase the volume of growing stock of Finnish forests due to increasing forest

growth (see e.g. Kellomäki et al., 2008). However, warming is also expected to increase certain risks to forests (e.g. Kellomäki et al., 2008, 2010; Peltola et al., 2010, Gregov et al., 2011; Lehtonen et al., 2016a, b). In the past few decades, wind storms have damaged a significant amount of timber and caused large economic and ecological losses in forestry from central to Northern Europe (Schelhaas et al., 2003; Gregow, 2011, 2013). In Finland, strong winds have damaged over 24 million $m^3$ of timber in different winter and summer storms since 2000 (e.g. Gregow, 2011; Zubizarreta-Gerendiain et al., 2016). During the

coming decades, the risk of wind damage to forests is expected to increase, although the frequency and severity of the storms may not increase. (Nikulin et al., 2011; Pryor et al., 2012) This increase is due to the shortening of the frozen soil period, which currently improves tree anchorage during the windiest season of the year from late autumn to early spring (Peltola et al., 1999a; Venäläinen et al., 2001; Kellomäki et al., 2010; Gregow et al., 2011).

In addition to the properties of wind (e.g. speed, direction, gustiness and their duration), the stand and site characteristics affect largely the vulnerability to wind damage (Peltola et al., 1999b; Gardiner et al., 2016). In Finnish conditions, mature stands adjacent to newly clear-cut areas or recently heavily thinned stands are especially vulnerable to wind damage (e.g., Laiho, 1987; Zubizarreta-Gerendiain et al., 2012). Risks to these forests may be decreased by proper forest management and planning for the spatial and temporal patterns of cuttings in forested areas (Tarp and Helles, 1997; Meilby et al., 2001; Zeng et al., 2004,

2007a; Heinonen et al., 2009; Zubizarreta-Gerendian et al., 2016). Several mechanistic models that have been built in recent decades allow the prediction of threshold wind speeds that can uproot or break trees under alternative forest stand configurations (e.g. Peltola et al., 1999b, 2010; Gardiner et al., 2000, 2008; Byrne and Mitchell, 2013; Seidl et al., 2014; Dupont et al., 2015). Consequently, based on these predicted threshold wind speeds it will be possible to predict the probability (and amount) of wind damage based on local wind characteristics if sufficient knowledge about the local wind climate is

available (e.g. Gardiner et al., 2008; Blennow et al., 2010; Zubizarreta-Gerendian et al., 2016).

An estimation of the frequency of extreme weather events, like extreme wind speeds, can be accomplished by utilizing extreme value analyses (EVA) methods, making possible to fit their statistical distribution (e.g., Gumbel, Frechet or Weibull distribution) into observations that offer the best estimate of the occurrence probability of the most extreme values of the

30 studied phenomena (e.g. Coles, 2001). The software package, Extremes Toolkit, developed by the National Center of Atmospheric Research (NCAR) is a widely used example of a tool that can be utilised to produce such a statistical distribution (Gilleland and Katz, 2011). For an accurate estimation of the probability of the occurrence of very extreme events with long return periods (e.g., 50 to 100 years), observations over many decades are needed. Additional difficulty to gain an accurate estimation of return levels of extreme wind speeds and wind gusts is caused by a lack of homogeneous wind observation time

series due to changes in the measuring conditions and instruments (Laapas and Venäläinen, 2016). One possibility of assessing the return levels of extreme wind speed in coarse resolution is to use reanalysed datasets (e.g., Dee et al., 2011), which are produced by assimilating all available observations in a systematic way. The benefit of these data sets is that they offer consistent spatial and temporal resolution over several decades (and hundreds of variables). Reanalysed data sets are also relatively straightforward to handle from a processing standpoint. Although the quality of this data varies from location to

location and from variable to variable, the scale of the magnitude of extreme wind for a coarse spatial scale can indeed be estimated based on them (e.g., Brönniman et al., 2012).





The high resolution spatial variation of extreme wind speed that is affected by topography and surface characteristics (e.g., Wieringa, 1986) can be considered by applying spatial statistical tools (e.g. Etienne et al., 2010; Jung and Schindler, 2015). Additionally, complex airflow models like WAsP (Mortensen, 2015) and WindSim (http://www.windsim.com/), typically applied for wind power potential predictions, can be used for this purpose. One example of exposure estimation is the Detailed

Aspect Method of Scoring (DAMS), which takes into account the local wind climate, elevation, aspect and topographic exposure (Quine and White, 1993; Hale et al., 2015). DAMS is used in the ForestGALES (http://www.forestry.gov.uk/forestgales) for estimating the probability of wind speeds that cause damage. GIS- based methods for mapping the most wind damage prone areas have also been introduced (e.g., Talkkari et al., 2000; Zeng et al., 2007b; Schindler et al., 2012; Ruel, 2002). A pragmatic and computationally very feasible approach to use to estimate the return levels

of extreme wind speeds for large geographical areas with very high spatial resolution is the wind multiplier approach. In this approach, regional wind speeds obtained, e.g., from the reanalysed data (or climate change scenario), are downscaled to local wind speeds to consider the local effects of land cover and topography (e.g. Jones et al., 2005; Cechet et al., 2012; Yang et al., 2014). By applying GIS-tools to detailed land-use data and digital elevation maps, it is also very straightforward to produce the required multipliers.

In this study, we evaluated the applicability of the wind multiplier approach for an estimation of the high-resolution (20 m) variability of extreme wind speeds in Finnish forested landscapes, employing CORINE- land use and high resolution digital elevation data. More specifically, wind multipliers were used to provide quantitative estimates of local wind conditions relative to the regional wind speeds in our 20 km x 20 km test area located in northern Finland and for 26 meteorological observing

stations with various surface characteristics in different parts of the country as well. The data processing was done using standard Python, QGIS, and ArcGis- software routines. The work was motivated by the fact that full-filling of the increasing needs of forest biomass for the growing forest-based bioeconomy will require also increasing wood harvesting intensity in thinning and final felling areas, thus potentially increasing the wind damage and other risks to these forests. Having reliable high-resolution information on extreme wind speeds expected over forested landscapes will enhance both forest management

and planning.

## 2 Material and methods

### 2.1 The wind multiplier approach

The wind multiplier approach used here follows the one presented in AS/NZS 1170.2 (2011) and Yang et al. (2014), where terrain properties are taken into account when assessing local maximum wind speeds (see Eq. 1). The return level of regional

maximum wind speed ($V_R$) in an open terrain at a 10 meter height is downscaled into site specific maximum wind speed ($V_{site}$) as follows:

$$V_{site} = V_R \times M_Z \times M_s \times M_h \times M_d \qquad (1)$$

Where the three wind multipliers used are the terrain/height multiplier ($M_z$), shielding multiplier ($M_s$) and topographic (hill-shape) multiplier ($M_h$). The impact of wind direction is taken into account using a fourth factor ($M_d$). In our study, we use a

20 m x 20 m grid, which is in line with the CORINE Land Cover 2012 dataset. It provides information on land cover and land use in 2012, and its changes from 2006 to 2012 (based on the European Gioland 2012 project). Our interest is forested landscape (including practically no buildings or other similar obstacles); therefore, the shielding factor was not considered. The return levels of winds speeds ($V_R$) were also defined separately for the eight cardinal and intercardinal wind directions. Thus, there was no need to calculate the direction multiplier ($M_d$).





### 2.2 Estimation of return level for regional maximum wind speeds

The regional scale return levels of maximum wind speeds were calculated using the ERA-Interim dataset (Dee et al., 2011) and the Generalized Extreme Value method (GEV) (e.g. Coles, 2001). This method estimated the 10-year return level of maximum wind speed as, for example, in inland Finland at below 12 ms$^{-1}$ and on open sea at even around 24 ms$^{-1}$ (Fig. 1). The

values are given as grid box averages, each covering an area of 0.75° × 0.75° and the time period used for the calculation of return levels covered years 1979-2015. The maximum wind speed dependence on wind direction was estimated by making the calculations wind direction wise (Fig. A1).

### 2.3 Estimation of the impact of terrain roughness on maximum wind speeds

In AS/NZS 1170.2 (2011) for elevations below 50 m, 1000 m fetch was used when the surface roughness impact was estimated.

In this study, we applied a somewhat different approach. First, each CORINE-land use class was interpreted to roughness lengths following the technique applied in the production of the Finnish Wind Atlas (Tammelin et al., 2013). When estimating the impacts of upwind conditions on wind speed in the location that was of interest, we used 500 m fetch to calculate the effective roughness ($z_{eff}$). As the conditions close to the place of interest have a larger importance than the values at a further distance, each 20 m grid cell did have a weighting factor *(we),* which was presumed to follow normal distribution (Eq. 2).

$$we_i = \frac{1}{\sigma\sqrt{2\pi}} e^{-\frac{1}{2}\left(\frac{x_i-\mu}{\sigma}\right)^2} \hspace{2cm} (2)$$

where σ is the variance defining the shape of distribution and in our case (150), *x* is the fetch, and μ is the location and in this case zero. Thus $z_{eff}$ was calculated as

$$z_{eff} = \sum_{i=1}^{25}(we_i \times z_{oi}) \hspace{2cm} (3)$$

where $z_{oi}$ is the surface roughness length of *i*th grid 20 m grid sell. The final step to calculate the surface roughness dependent

multiplier ($M_z$) was to use the estimates given in Tables 3.2 and 3.3 by Yang et al. (2014). This step led to an estimate given in Eq. 4.

$$M_z = -0.056\ln(z_{eff}) + 0.7715 \hspace{2cm} (4)$$

The multiplier were defined for eight directions (cardinal and intercardinal), using the GDAL raster utility programs.

In ERA-Interim analyses, a roughness length for each grid cell is presumed. To normalize the roughness length of the ERA-Interim data into a reference roughness, we multiplied the ERA-Interim wind speed values by $1/M_z$ (Eq. 4), using the ERA-Interim grid cell roughness length as the value of $z_{eff}$.

The values of $Z_0$, $Z_{eff}$ and $M_z$ in the case of sharp roughness change between forest and lake are demonstrated in Fig. 2. When

the wind comes from an open lake ($z_0$=0.0004 m) to dense forest ($z_0$=1.4 m), then the multiplier $M_z$ changes from 1.21 to 0.75 within a distance of 300 m and to 0.80 within a distance of 80 m. The change is very rapid, demonstrating the strong slowing of wind speed within a dense forest during the first tens of meters. This rapid change is demonstrated in the case when the most vulnerable forest edges are studied; the wind throw risk is largest within approximately 20-30 m from the upwind edge of a clear cut area (e.g. Peltola et al. 1999b). The acceleration of wind speed from forest to open water surface is not as rapid

as the slowing; the change of $M_z$ from the minimum value of 0.75 to maximum takes about 500 m (Fig. 2). This rate of acceleration is quite close to the values introduced by Venäläinen et al. (1998). The values of $Z_{eff}$ and $M_z$ for the Pyhtätunturi-fell area for northwesterly winds are given in Fig. A3.

### 2.4 Estimation of the impact of topography on maximum wind speeds

The topographic multiplier $M_h$ was taken as the larger of the two estimates $M_{h1}$ and $M_{h2}$ (Eq. 5).

$$M_{h1} = 0.4343 \times \ln(H_{eff}) \hspace{1cm} \text{for } M_{h1} < 1, M_{h1} = 1 \hspace{1cm} (5a)$$





$$M_{h2} = 0.913 \times e^{0.0008 H_{msl}} \tag{5b}$$

$M_{h1}$ simulates the impact of small scale topographic variation that is typical in Finland. $H_{eff}$ is calculated as the difference between the place of interest and the median elevation of 1000 m distance upwind from the location of interest (see Fig. 3).

The logarithmic shape follows that of logarithmic wind law in the case of surface roughness one meter that is typical for a forest and wind speed 15 ms⁻¹ at an elevation of 10 m. The other multiplier $M_{h2}$ simulates the general increase of wind speed as a function of elevation. The shape of Eq. 5b is based on an estimate of the dependence of a 50- year return level of maximum wind speed on elevation, defined by using wind measurements made at observing stations located at different elevations (not published). Finland is a rather flat country, and most of the country is located below an elevation of 200 meters above sea

level. Multiplier $M_{h2}$ is thus larger than $M_{h1}$ at only very rare locations in the entire country. The topographic wind multiplier $M_h$ was calculated using the digital elevation data obtained from the Finnish National Data Survey. The data was 25 m spatial resolution raster data re-sampled to 20 m resolution. The data was first smoothed by replacing each pixel with the average of its $3 \times 3$ neighborhood, and done to filter out the very small scale noisy features the data might contain. The processing was done by utilizing the R package 'raster'. $H_{eff}$ was then calculated in the eight directions of the wind, using Python and the

GDAL routine.

An example of the change of topographic multiplier in the case of a transection reaching over the roughly 500 m high Pyhätunturi (Fig. 5) fell in Northern Finland in case of north-westerly wind is given in Fig. 4 and Fig. A4. As this place is located at a relatively high elevation, the purely on elevation dependent $M_{h2}$ dominates, and only in the case of a steep hill

slope around the location interval 11655 m-13405 m (Fig. 5) does the multiplier $M_{h1}$ get larger values than $M_{h2}$. The approach used in this study is also simpler than the one described in AS/NZS 1170.2 (2011). Still, as the terrain in Finland is relatively flat, the main impact of these relatively small scale topographic variations can be taken into account even with the schema utilized here.

## 2.5 Verification tests

The first verification tests were done by utilizing wind measurements made at 26 observing stations in Finland, and of these, 23 stations belong to the observation network maintained by the Finnish Meteorological Institute (FMI) and represent conditions ranging from open sea to agricultural land, forests, and open hill areas. More detailed analyses were made in Northern Finland (67.02204° lat, 27.2184° lon) for the Pyhätunturi –fell test area, with an elevation range of 148 – 526 m above sea level (Fig 5, Fig. A2). In addition to the forests, other terrain types included open tundra, agricultural fields, lakes,

and ski slopes. The test area had both larger topographic variation and spatial variation of wind speeds than the typical Finnish landscape and in that sense represented more challenging conditions than those expected in most of the rest of the country. This Pyhätunturi –fell test area was also used in the EU- funded MOWIE project, where three 10-m tall wind-measuring masts were installed at a range of elevations above the mean sea level: 470 m (MM1), 419 m (MM2) and 408 m (MM3). In addition, there was a permanent observing mast (FMI station number 7708) on the telecommunication mast (TM) at an elevation of 61

35  m above the surface at the top of the hill (see Fig. 5). All wind speed measurements were corrected to 10 meter high values by applying the logarithmic wind law. Wind climate simulations for this same area have earlier been made utilizing the Karlsruhe Mesoscale Model (KAMM) and WAsP by Frank et al. (1999).

Based on the measurements made at the observing stations, 10 year- return levels of maximum wind speeds were calculated

for each location and compared with return period values obtained for the station locations using the wind multiplier approach and the ERA-Interim maximum wind speed estimates. For stations MM1, MM2 and MM3, there was only two years of data available, a short period to estimate even 10-year return levels. Therefore, to have the extreme value analysis be as robust as




possible, for these stations, we applied the Block Maxima approach (e.g., Coles, 2001) to the monthly maximum values, using the R package extRemes (Gilleland and Katz, 2016). For most of the other station locations, the data used for the extreme value analyses covered the years 1979-2015.

For the Pyhätunturi –fell area, we also compared a spatial variation of high wind speed as simulated by the WAsP- package with a wind multiplier downscaled wind. The area was slightly smaller (Fig. 5) due to the availability of terrain information needed for a WAsP simulation. In the WAsP simulation, the geogstrophic wind speed was expected to be 39 ms$^{-1}$ from north-west. This geogstrophic wind speed leads to approximately 26 ms$^{-1}$ winds at the top of the Pyhätunturi –fell, which is roughly the 10-year return level maximum wind speed (see Table 1). For a comparable wind multiplier downscaling, the coarse scale

north-westerly wind 12.7 ms$^{-1}$ was used as the basis in the calculation. In this way the maximum wind speed was the same in both simulations.

## 3 Results

### 3.1 Comparison of measurement-based return levels to those based on a wind multiplier approach

A comparison of the ERA-Interim and wind multiplier- based assessment of 10-year return levels of wind speed to the estimates

based on measurements for the test locations (Table 1, Fig. 6) revealed that for these locations and representing different kinds of terrain and elevations, the wind multiplier approach improved the local wind speed return level estimates remarkably ($R^2$ = 0.766). There was also no clear bias in the estimates, the mean difference being 0.13 ms$^{-1}$ and the RMSE error 3.31 ms$^{-1}$. The largest differences were found in the case of Station No. 9004, which is located at an elevation of 1004 m above sea level, i.e. almost at the highest point in Finland. The anemometer at this station is also located at the edge of a steep slope, thus leading

to a high topographic multiplier value.

At the four Pyhätunturi -fell stations, the wind multiplier estimates were close to the measurement- based estimates with the exception of Station MM1. The estimate based on measurements made at MM1 (29.6 ms$^{-1}$) was almost 7 ms$^{-1}$ higher than the return level estimate calculated for the telecom mast at the same height (the TM value at an elevation of 61 m was roughly the

same as the value at Station MM1). The difference between MM1 and MM2 was about 3 ms$^{-1}$. The return level estimates for Stations MM1, MM2 and MM3 were based on two years of measurements and led also to a high degree of uncertainty. For example, for Station MM1, the estimated 95% confidence levels were 23.1 ms$^{-1}$… 36.1 ms$^{-1}$. The corresponding estimate of the telecom tower based on 10 years of measurements was more robust with 95% confidence levels, i.e., 21.8 ms$^{-1}$… 24.7 ms$^{-1}$.

### 3.2 Spatial variation of maximum wind speeds

The spatial variation of 10-year return levels of wind speeds within the roughly 4000 km$^2$ Pyhätunturi test area was large. The lowest values for the 10-year return level were around 9.2 ms$^{-1}$ and the highest on top of the Pyhätunturi-fell were approximately 26.5 ms$^{-1}$ (Fig. 7). A crude approximation indicates that mean 10 min wind speed exceeding 12 ms$^{-1}$ can uproot or break a tree during unfrozen soil conditions (Peltola et al., 1999b; Zubizarreta-Gerendiain et al., 2012). When we looked at

the spatial variation of a 10-year return level of wind speed inside the test area, we can see areas having wind speeds higher than the threshold of 12 ms$^{-1}$ found on local topographic formations, at the edges of open terrain, and at high elevation locations. At a total 23.8% of grid squares, the 10-year return level wind speed reaches the threshold and if we look only at the forested area, then we end up with 22.8 %. This statistic means that approximately 20% of the area is exposed to wind speeds that can lead to forest damage. The exact value depends, however, on several factors including tree and stand characteristics.



In a qualitative comparison, the wind multiplier approach and a WAsP simulation produced the same dominant features of spatial variation of maximum wind speed; maximum values were found at treeless fell-top areas (Fig. 8). One interesting feature was the case of the WAsP- simulation for the acceleration of wind at the forest-lake edge; it was immediate, and so was the deceleration on the opposite shore. In such a case of wind multiplier simulation, the impact of roughness change is

5 reflected a longer distance, as can be seen in the case of the Lake Pyhäjärvi. On top of the fell, the wind speed was adjusted to approximately the same 26 ms[-1]. On the lee side of the top of the fell, the wind multiplier simulations indicated a more rapid deceleration of wind speed than WAsP, while on the side to windward, the wind multiplier gave higher wind speeds. With no proper measurements, we could not decide which reflected the real conditions better. In the case of canyons like Pyhäkuru and Pikkukuru (Fig. 8) WAsP is more capable of predicting higher wind speed values than the wind multiplier and obviously

reflect the prevailing conditions better. For most of the lower elevation areas, the difference between the two simulations was small, and with these input wind speeds, the prevailing difference is on the scale of 1 ms[-1]; wind multiplier giving systematically higher wind speeds (Fig. A5). By scaling wind multiplier input wind speed lower the bias could have been adjusted to zero.

## 4 Discussion and conclusions

### 4.1 Reliability of tested method

The wind multiplier method has been used earlier to estimate the design values of buildings and other constructions (AS/NZS 1170.2, 2011) and assessment of wind damage risk (Yang et al., 2014). Based on our study, the wind multiplier method is very capable of identifying the locations having the highest extreme wind speeds in Finnish conditions. This is true despite the fact that this approach is much simpler than the dynamical models. The wind multiplier approach is also easily transferable to any location with needed terrain information and is an interesting and easily applicable alternative to use to assess the exposure of

terrain.

How precise each grid square estimate is depends on several external factors. First, we must have an estimate of the coarse scale return levels of the extreme wind speeds. Reanalysed data gives such a coarse estimate. If the reanalysed data is compared to in situ measurements in certain wind storm event, it is easy to find large differences between them. As well, the return levels

of wind speed calculated using ERA-Interim grid values can be quite different from the value based on point measurements, but downscaling the grid value to the point using the wind multiplier approach improves the estimate substantially, as we demonstrated in Fig. 6. It is also good to remember that although the wind measurements made at meteorological stations can go through several quality control steps, they still may contain erroneous values. As well, the measured values used here have not been homogenized, and e.g., changes in anemometer location and terrain properties may influence the values. In that sense

the return periods based on measured values (Table 1, Fig. 6) contain several uncertainties that are wise to remember when the comparison is fully valued.

The simple visualization and comparison of the spatial variation in wind speed at Pyhätunturi –fell was done by applying WAsP and, on the other hand, by applying wind multipliers. These demonstrate that the main features of spatial variation of

35 an extreme wind field produced by these two different methods are very similar. A profound analysis on the exact accuracy of the simulations is not possible, however, based on the available measurements; it would require much more detailed and reliable wind measurement data. However, by fine- tuning the wind multipliers, it is possible to achieve results that are closer to WAsP simulation. Pyhätunturi is not a typical Finnish forested landscape due to its high topographic variation. In those parts of the test area that exemplify a more typical landscape with only relatively small topographic features these two methods

give quite similar results. It is also good to remember, that as we are summarizing all wind directions (Fig. 7) the importance of lee -side wind simulation accuracy is not as crucial as having accuracy for the windward size having the highest wind speeds.



The wind multiplier method itself is also relatively easy to apply. The calculation of surface roughness and topographic multipliers can be done using routine GIS tools, and these calculations can be done for large areas like, e.g., the whole country. Similarly, this method could be used to assess the risks to forests that are related to forest management and planning with
relatively little extra effort. Further, climate change impact assessments can be done with high spatial resolution when the return levels of maximum wind speed are calculated using climate scenarios instead of only reanalyzed data.

One challenge of the method is the accuracy of surface roughness information in the CORINE-dataset; it is updated approximately every six years and thus does not represent real-time land use conditions for all locations. For example, forest
clear-cutting changes the roughness conditions very dramatically. Thinning affects it less. More frequent updates to surface conditions could be obtained from satellite measurements. As an example, the European Space Agency's (ESA) satellite Sentinel-2 ([http://www.esa.int/Our_Activities/Observing_the_Earth/Copernicus/Sentinel-2](http://www.esa.int/Our_Activities/Observing_the_Earth/Copernicus/Sentinel-2)) is producing high spatial resolution, at best 10 m, data describing the earth surface properties. Because of the development of satellite-measured data handling methods, the data can provide new possibilities for updating the surface state with a higher frequency than, e.g., the
CORINE-data is updated. Use of up- to -date airborne laser scanning data, if available (e.g., Kotivuori et al., 2016), can also offer a viable means to provide very detailed information on forest properties and thus also offer information on surface roughness conditions.

### 4.2 Conclusions

The rapidly growing, forest-based bioeconomy calls for increasing wood harvesting intensity, which means an increase in
thinning and a final felling area. This circumstance will increase the wind damage risks especially at the upwind edges of new cleared felling areas and thinned stands that have not yet been acclimated to increasing wind loading. Thus proper risk assessment is a clear pre-condition for a sustainable forest-based bioeconomy. This study demonstrates a useful tool to use for forest management and planning.

The tested wind multiplier method is very capable of identifying the locations (at high-resolution) having the highest extreme wind speeds and could well support the precise assessment of wind damage risks to forests. It can also be used to provide needed wind climate information for wind damage risk model calculations, thus making it possible to estimate the probability of predicted threshold wind speeds for wind damage, and consequently the probability (and amount) of wind damage under certain forest stand configurations. Accurate estimation of the spatial variation of the return levels of extreme wind speed with
very high spatial resolution over the whole country or even over larger areas like Fennoscandia are possible in the future using this approach. A high resolution estimation of climate change impacts on forest wind damage risks is also feasible using this approach.

**Acknowledgements**
This work was supported by the Strategic Research Project FORBIO, which is funded by the Academy of Finland.

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





Table 1. 10-year return levels of maximum wind speed as estimated directly from ERA-Interim data, downscaled to station locations using the wind multiplier approach, and calculated based on measurements for the station locations given in Fig.5. The other variables included are the wind multipliers ($M_z$, $M_{h1}$, $M_{h2}$), surface roughness, station elevation above sea level ($H_{msl}$) and local topography ($H_{eff}$) and estimated direction of strongest winds (Dir.).

| Station number | ERA Interim | Wind multiplier | Measure-ments | $z_{eff}$ | $M_z$ | $H_{msl}$ | $H_{eff}$ | $M_{h1}$ | $M_{h2}$ | Dir. |
|---|---|---|---|---|---|---|---|---|---|---|
| 1 | 20.0 | 18.4 | 15.6 | 0.07 | 0.920 | 1 | 10 | 1.000 | 0.914 | s |
| 2 | 19.7 | 17.3 | 21.0 | 0.16 | 0.874 | 9 | 10 | 1.000 | 0.920 | sw |
| 3 | 20.0 | 20.5 | 21.0 | 0.01 | 1.029 | 8 | 10 | 1.000 | 0.919 | s |
| 11 | 20.0 | 16.9 | 17.3 | 0.27 | 0.845 | 11 | 10 | 1.000 | 0.921 | sw |
| 101 | 20.1 | 22.6 | 21.8 | 0.2 | 0.862 | 5 | 20 | 1.301 | 0.917 | nw |
| 103 | 16.9 | 13.8 | 14.0 | 0.44 | 0.817 | 6 | 10 | 1.000 | 0.917 | s |
| 301 | 17.7 | 17.6 | 15.9 | 0.02 | 0.991 | 51 | 10 | 1.000 | 0.951 | sw |
| 302 | 19.1 | 18.2 | 20.1 | 0.5 | 0.810 | 10 | 15 | 1.176 | 0.920 | w |
| 1201 | 16.0 | 14.4 | 12.4 | 0.1 | 0.900 | 104 | 10 | 1.000 | 0.992 | sw |
| 2710 | 13.7 | 12.5 | 12.8 | 0.08 | 0.913 | 93 | 10 | 1.000 | 0.984 | s |
| 3601 | 13.8 | 13.7 | 14.1 | 0.02 | 0.991 | 99 | 10 | 1.000 | 0.988 | nw |
| 3801 | 13.6 | 13.2 | 12.1 | 0.03 | 0.968 | 121 | 10 | 1.000 | 1.006 | se |
| 6801 | 13.9 | 15.2 | 12.8 | 0.03 | 0.968 | 264 | 10 | 1.000 | 1.128 | w |
| 7708(TM) | 12.3 | 22.0 | 22.6 | 0.18 | 0.868 | 491 | 116 | 2.064 | 1.352 | e |
| 7708 (MM1) | 12.3 | 22.3 | 29.6 | 0.19 | 0.865 | 470 | 130 | 2.100 | 1.330 | e |
| 7708 (MM2) | 12.3 | 24.4 | 26.2 | 0.1 | 0.900 | 419 | 160 | 2.200 | 1.277 | e |
| 7708 (MM3) | 12.3 | 22.0 | 22.2 | 0.05 | 0.939 | 408 | 80 | 1.900 | 1.265 | e |
| 8307 | 14.4 | 30.9 | 34.8 | 0.076 | 0.916 | 760 | 220 | 2.342 | 1.677 | sw |
| 8308 | 14.4 | 25.8 | 24.7 | 0.049 | 0.940 | 565 | 80 | 1.903 | 1.435 | ne |
| 8312 | 14.4 | 17.4 | 15.0 | 1.39 | 0.753 | 347 | 40 | 1.602 | 1.205 | sw |
| 8601 | 13.5 | 17.1 | 18.7 | 0.0004 | 1.210 | 240 | 11 | 1.041 | 1.106 | n |
| 8607 | 14.1 | 24.7 | 23.2 | 0.128 | 0.887 | 437 | 97 | 1.987 | 1.295 | sw |
| 9003 | 19.9 | 21.8 | 19.7 | 0.44 | 0.817 | 480 | 10 | 1.000 | 1.340 | s |
| 9004 | 19.9 | 45.6 | 33.9 | 0.06 | 0.929 | 1003 | 294 | 2.468 | 2.037 | sw |
| 9603 | 18.7 | 22.1 | 21.0 | 0.26 | 0.847 | 107 | 25 | 1.398 | 0.995 | nw |
| 9705 | 15.1 | 12.2 | 16.5 | 0.53 | 0.807 | 121 | 10 | 1.000 | 1.006 | nw |





**Figure 1: 10 –year return level of maximum wind speed calculated using ERA Interim 1979-2015 data and the GEV-analysing approach.**





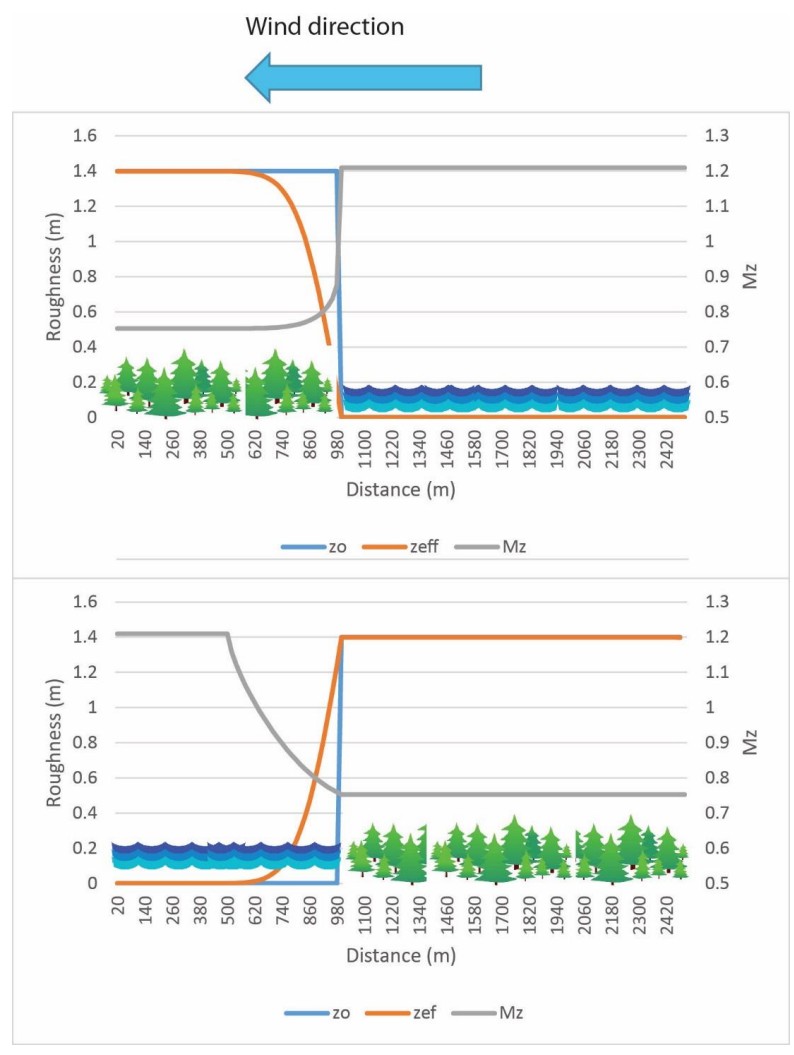

**Figure 2: The value of surface roughness lengths $z_0$ and $z_{eff}$ and the surface roughness dependent wind multiplier ($M_z$) in cases where wind is from lake to forest (upper figure) and from forest to lake (lower figure).**



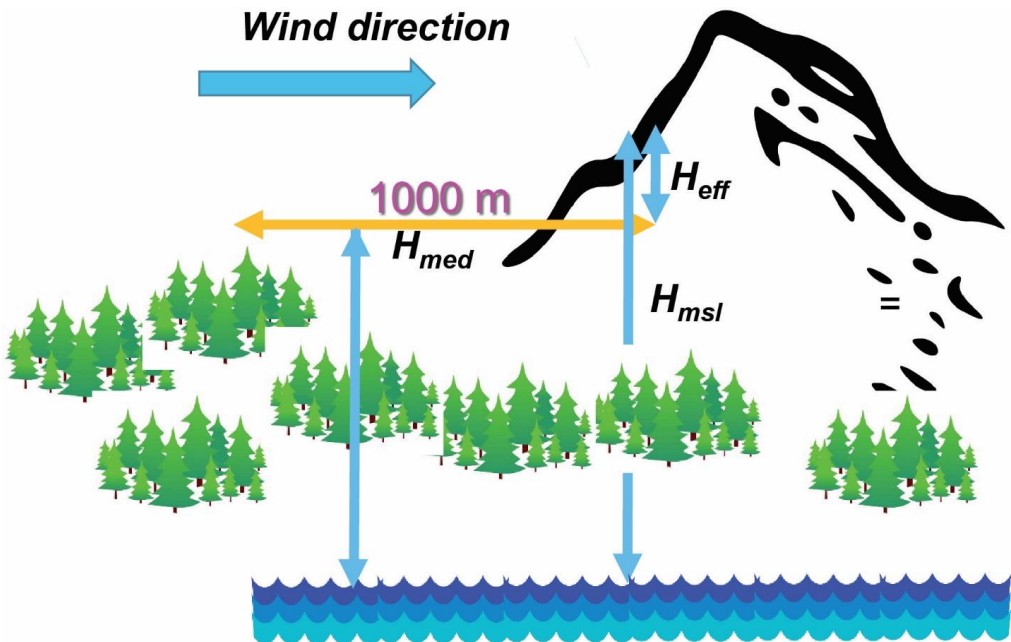

**Figure 3: A visualization of the calculation of the topographic multiplier M_h.**





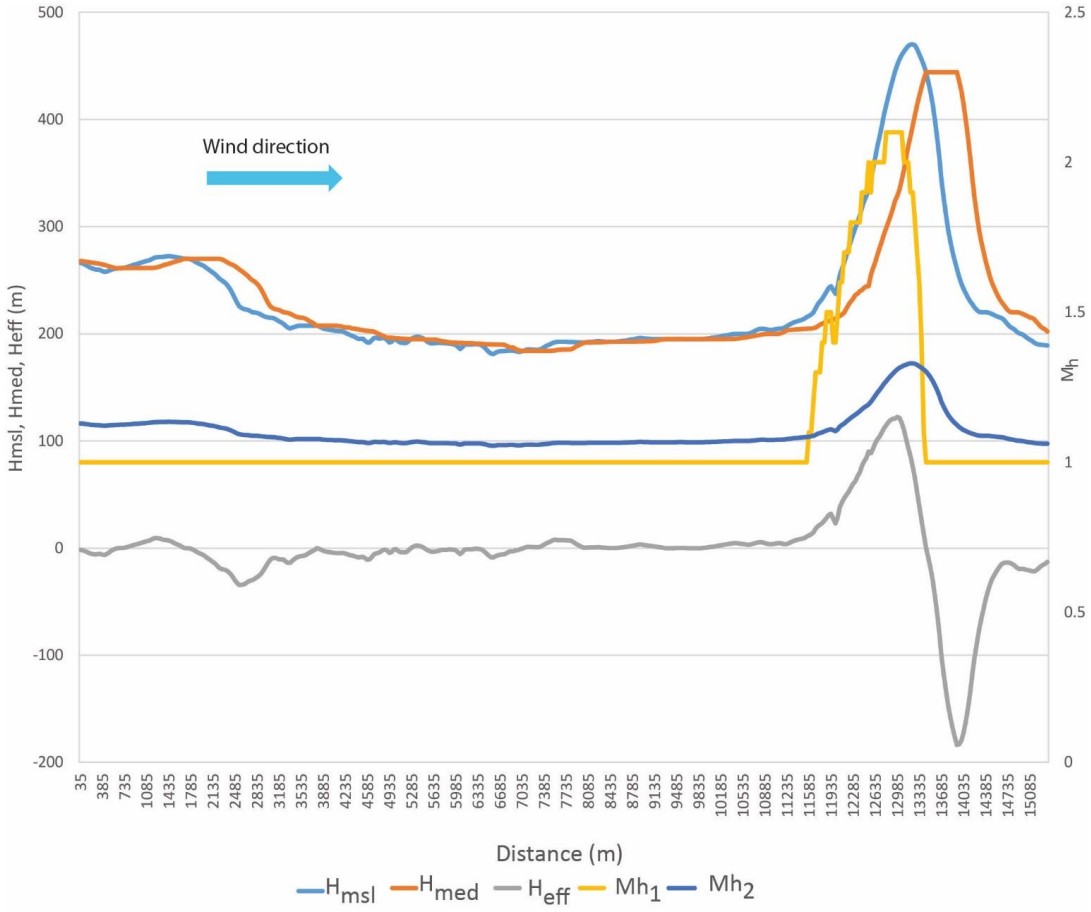

**Figure 4: Variations of elevation (Hmsl), median elevation (Hmed), and the effective elevation (Heff) (left axis) and topographic multipliers Mh1 and Mh2 (Eq. 5, Fig. 3) (right axis) along a transection from northwest to southeast (the black line in Fig. 5) crossing the Pyhätunturi- fell in Northern Finland.**





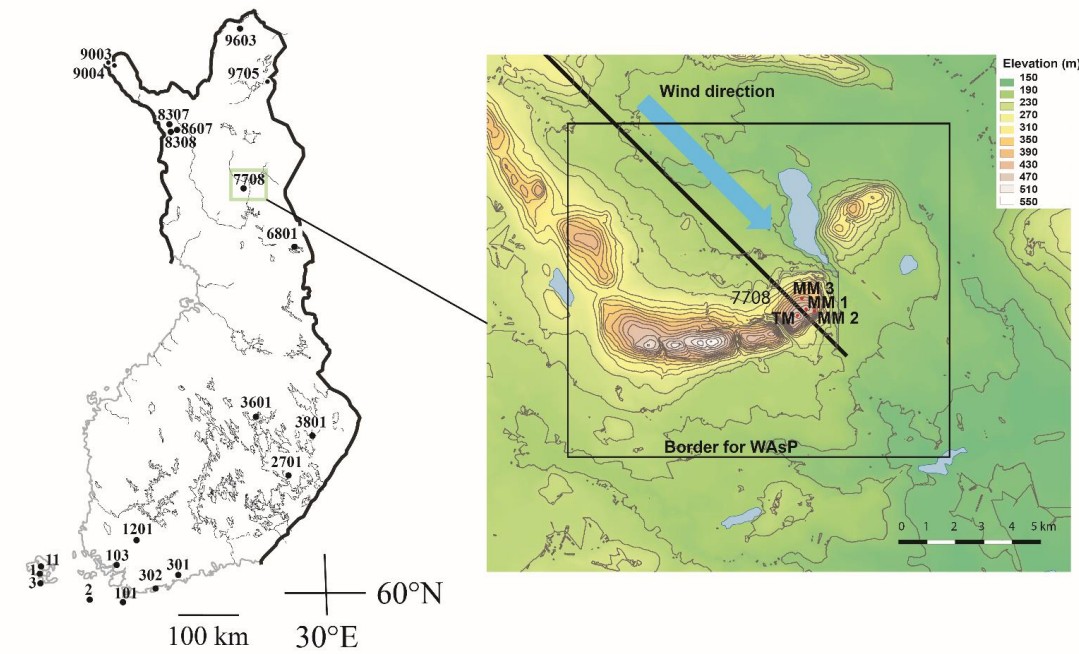

**Figure 5: The meteorological stations used in the analyses (see Table 1, Fig. 6) and the topography of the Pyhätunturi -area located in Northern Finland. The black northwest direction line in the Pyhätunturi figure indicates the transection analysed in Fig 4; the black square indicates the border of the WAsP -simulation.**




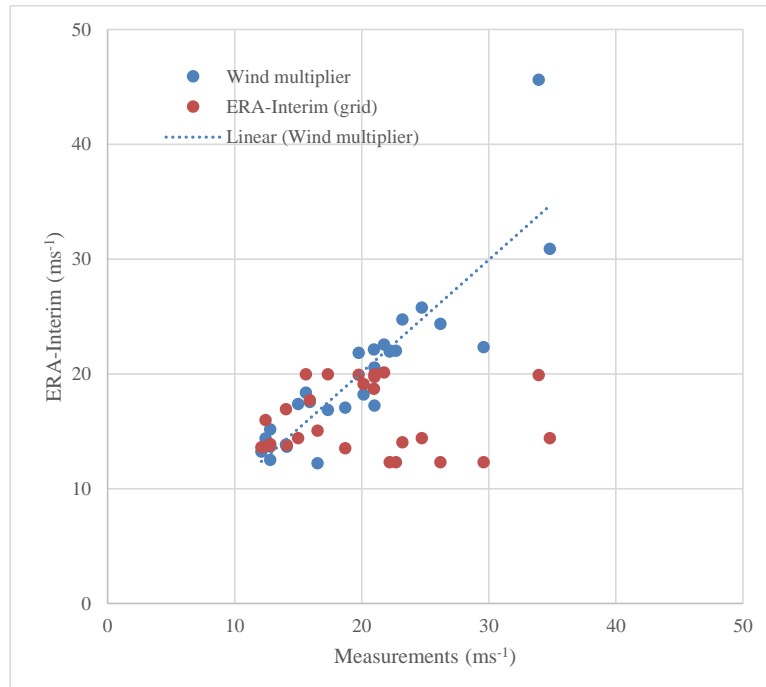

**Figure 6: Comparison of 10-year return levels of maximum wind speeds, as calculated, based on observations and by utilizing the wind multiplier method (Eq. 1) and the ERA-Interim dataset for the 26 measuring sites (Fig. 5). Return levels taken directly from the ERA-Interim dataset with no wind multiplier correction (grid) are included in the visualization.**





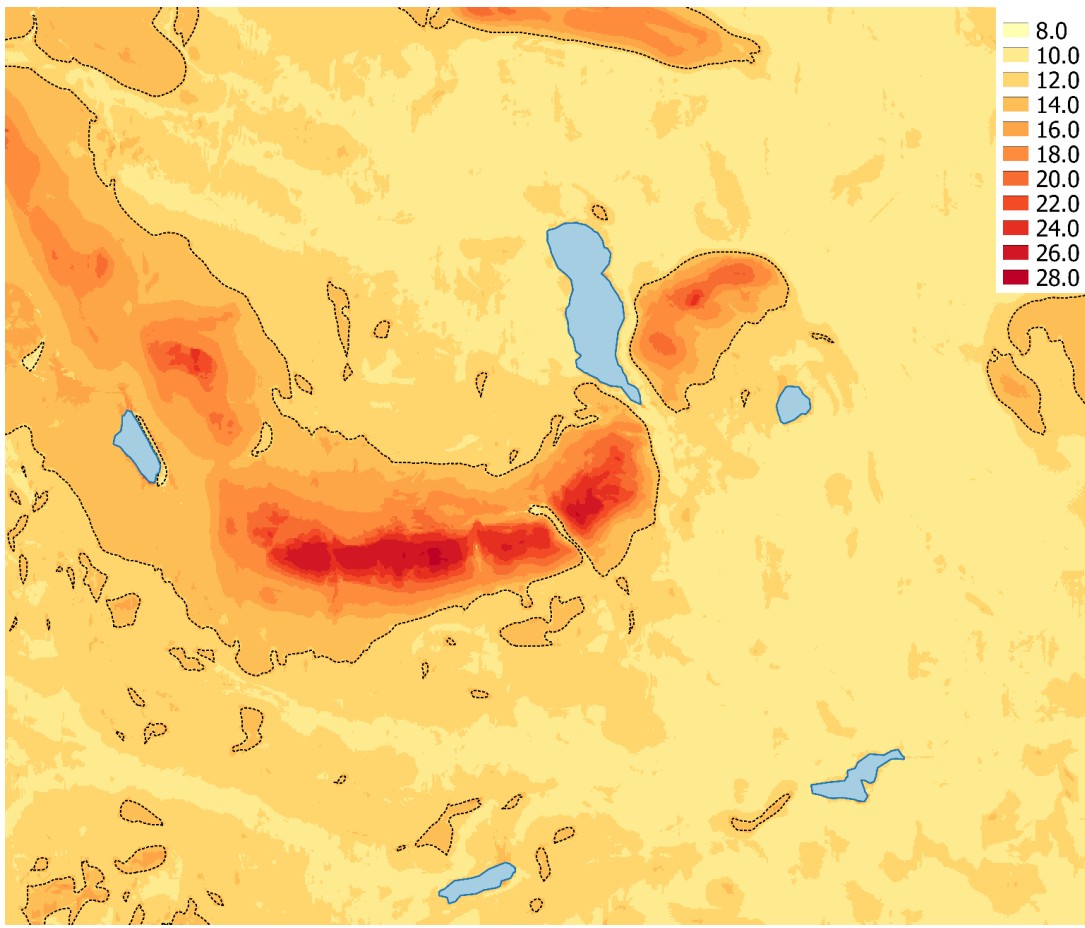

**Figure 7: 10-year return levels of maximum wind speed (A) calculated using the wind multiplier method (Eq. 1) and the ERA-Interim dataset for the Pyhätunturi test area (Fig. 5). The values where wind speed exceeded 12 ms⁻¹ are indicated by a black dotted line.**





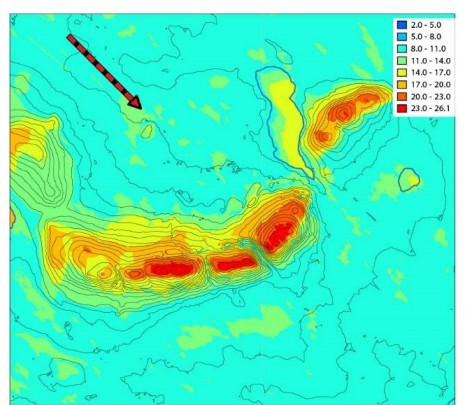    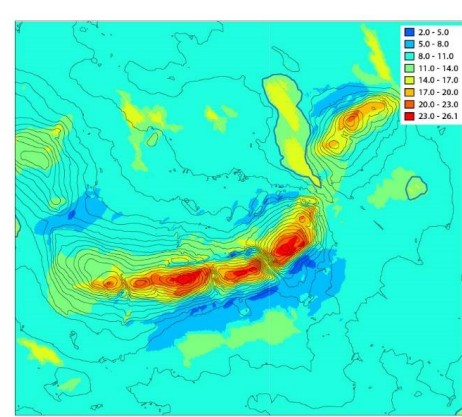

Wind multiplier                                WAsP

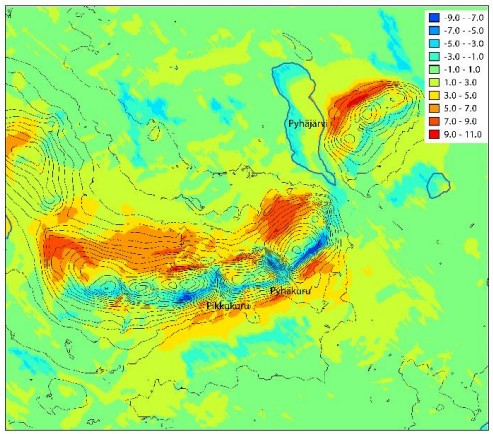

5                                Wind multiplier - WAsP

**Figure 8: Comparison of the spatial variation of wind speed as estimated, using the wind multiplier approach, calculated using the WAsP –programme. The last figure depicts the difference between the two methods. Wind direction is from the northwest and in the case of the wind-multiplier it is 12.7 ms$^{-1}$. For the WAsP simulation a geostrophic north-westerly wind of 39.2 ms$^{-1}$ was assumed.**

.



APPENDIX

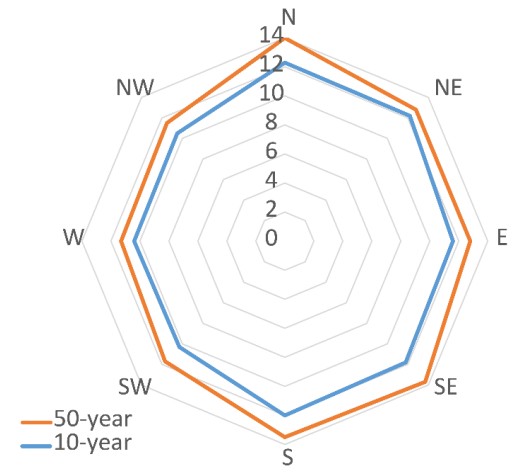
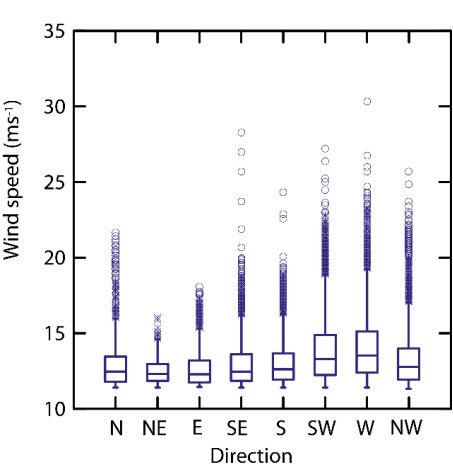

**Figure A1. Wind speed distribution for the 10- and 50-year year return levels of 10 m height maximum wind speed as estimated from the ERA-Interim 1979-2014 dataset for the Pyhätunturi-fell grid box. Box-plot depicting measured 10-minute wind speed values at the Pyhätunturi telemast station during years 1997-2016. The values measured at an elevation of 61 m were corrected to represent 10 m by applying logarithmic wind law. Only values that are 11.4 ms$^{-1}$ (corresponding value 15 ms$^{-1}$ at elevation of 61 m)**
10 **are included into the analyses (right panel).**

30





LEGEND

■ Artificial surfaces

■ Agricultural areas

■ Forests

■ Shrub and/or herbaceous vegetation associations

■ Open spaces with little or no vegetation

■ Inland wetlands

■ Inland waters

**Figure A2. Land-use map for the Pyhätunturi-fell area based on the CORINE dataset.**



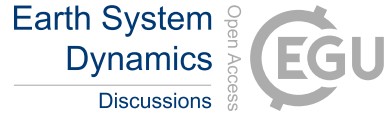

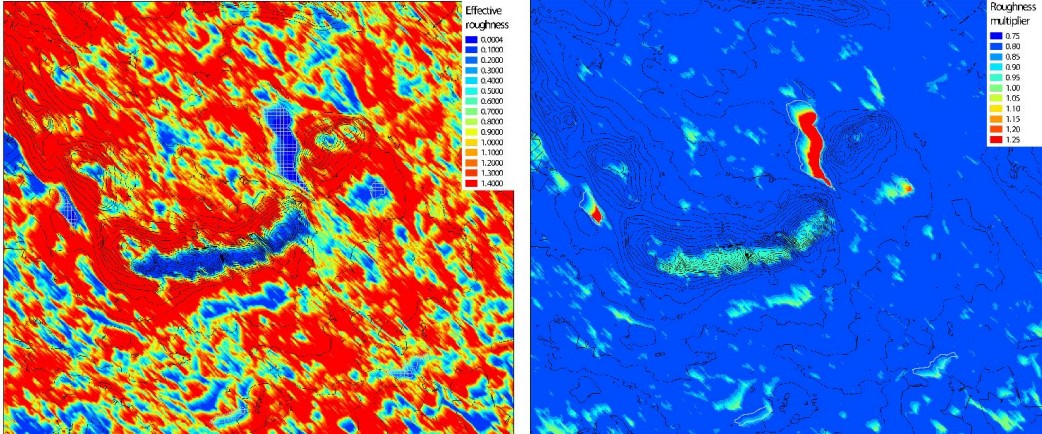

**Figure A3. Effective roughness (zeff, Eq. 4) and roughness dependent wind multiplier (Mz) calculated for the Pyhätunturi-fell area for northwesterly winds.**



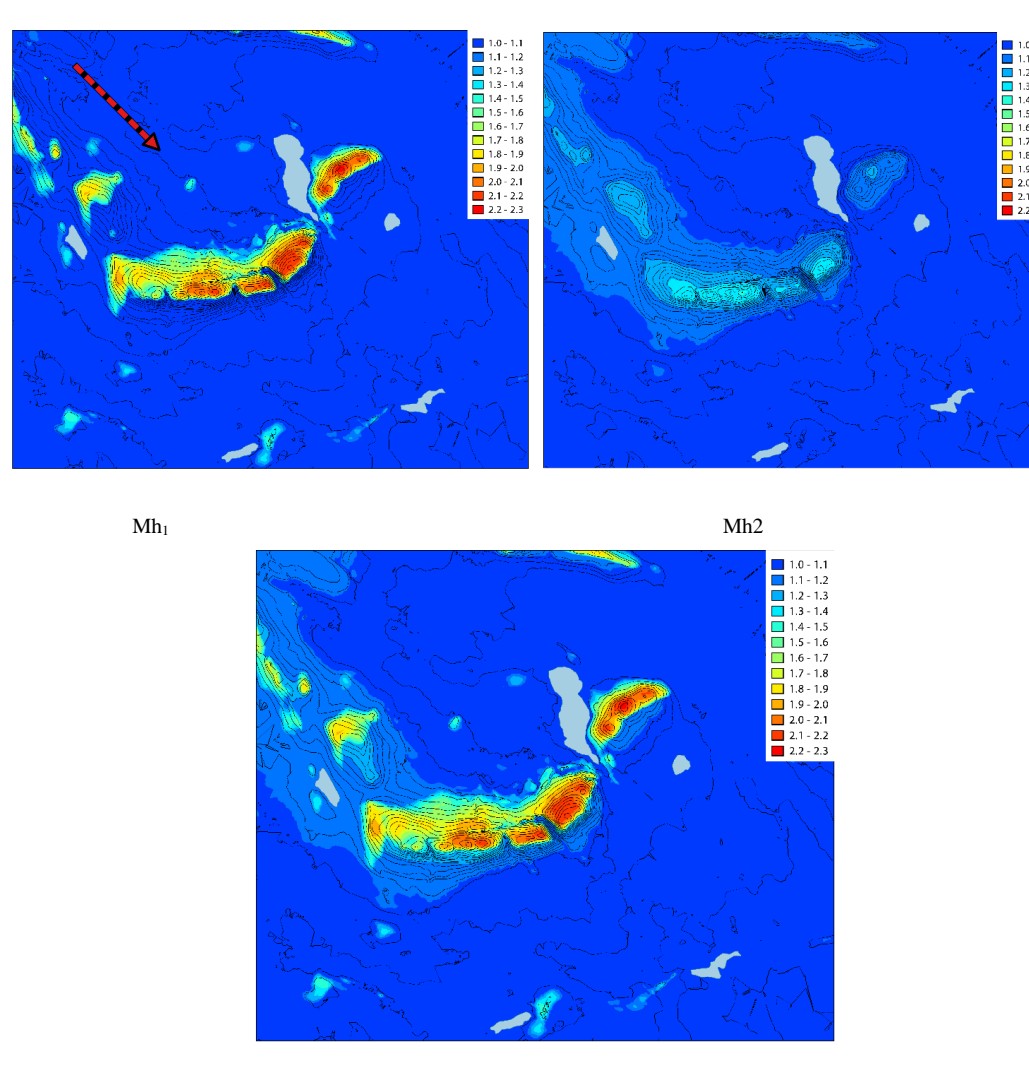

Mh₁                                    Mh2

5                        Mh_final

**Figure A4. Topographic wind multipliers (Eq. 6) calculated for the Pyhätunturi-fell area for northwesterly winds.**




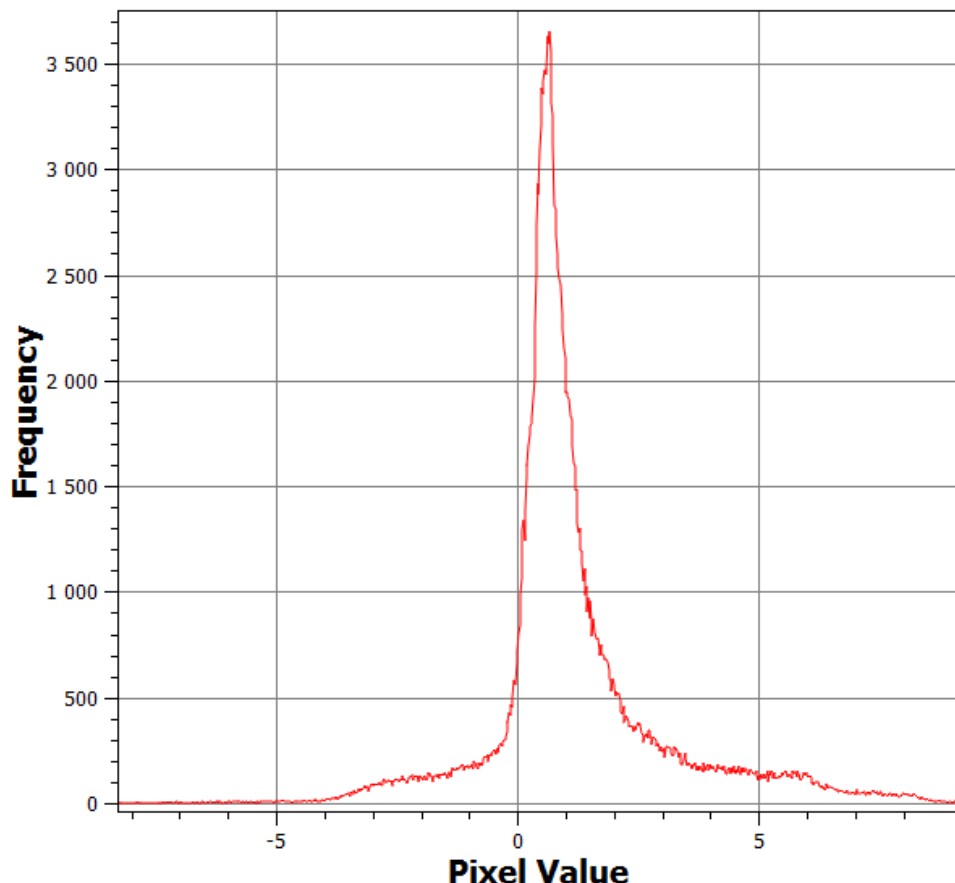

**Figure A5: Histogram depicting the difference between the wind speed as estimated using the wind multiplier approach and as calculated using the WAsP –programme (see Fig. 8).**