# Peer review of "Estimation of the high spatial resolution variability of extreme wind speeds for forestry applications Ari K. Venäläinen1, Mikko O. Laapas1, Pentti I. Pirinen1, Matti Horttanainen1, Reijo Hyvönen1, Ilari Lehtonen1, Päivi Junila1<"

_Earth System Dynamics, 2017_

## Referee Comment (RC1) · Anonymous Referee #1 · 9 Feb 2017

The paper tests a relatively simple approach to characterize high wind speeds across large regions. Even though the paper does not bring new knowledge about wind behaviour, I believe it is worth publishing because it tests in a rigorous way an innovative approach to incorporate wind climate in managing windthrow risk.

The study context is clearly stated. Managing windthrow risk at the local scale requires one to have an estimate of the recurrence of strong winds on a given site. Some wind circulation models can provide that kind of estimates but can hardly be used in an operational context. Hence, the interest for alternatives is great.

The methods are rigorous and seem to have been applied correctly. However, some additional details may be helpful. The proposed approach has been compared to Wasp, a model that has been used in many previous windthrow studies. Even though this is not a validation, it shows that, in most situations for the region, both approaches compare quite well. Comparisons with weather station data clearly demonstrate that the method is a major improvement over using regional estimates of wind speed.

Here are some minor comments: p. 4, l. 14: the rationale behind the weighting factor should be explained p. 6, l. 33-34: the authors use 12 m/s as a wind speed likely to cause damage. Given the availability of the HWind model, it would seem interesting to provide examples of stands that would be vulnerable to such wind speeds. The same applies at l. 39. I believe the paper could be strengthened if this section was somewhat expanded. p. 7, l. 28-29. The authors point out some potential imprecisions of weather station data. It would be interesting to know to what extent this problem is present in the data base. p. 13: please provide units with column headings p. 19: the title of the y axis should be changed since it represents 10 year return levels of maximum wind speed for two different approaches

p. 22: it is mentioned that the figure includes only values > 11.4 m/s. Do you mean that the graphs were truncated at this value or that the whole statistics did not consider lower values?

---

## Referee Comment (RC2) · Anonymous Referee #2 · 21 Apr 2017

General

The manuscript provides a framework in assessing the damage potential of extreme wind speeds on Finish forests applying a geospatial remapping technique on the ERA40-reanalysis. Results are compared to the original re-analysis output and on in-situ meteorological observations to assess the skill of the approach for a local test area. In general the manuscript is well written and structured, the area of research fits into the scope of ESD and also the topic is important in the perspective of interdisci-plinary research, supporting the publication of the article in ESD. However, for final pub-

lication the authors should include some additional information on other mechanisms potentially affecting forests and re-visit and explain some of their statistical approaches including uncertainties, e.g. that the length of their observational data might be critical in assessing decadal return periods of wind extremes.

Specific

Title: The title includes a bit more promises than actually addressed in the body of the manuscript – for risk management an assessment of potential future changes is necessary. Investigations about future scenarios are however not addressed in this study. Therefore I suggest to modify/correct the title for this missing part of the analysis.

Abstract: The abstract is very extensive – At the end of the review the authors find a suggestion for a shortened version with focus on the very background and the most important findings.

1 Introduction

p1. l. 5: could you please add a few examples from the papers you cite which specific risks are impacting on the forests in Finland. p2. l. 35ff: The authors should add one or two sentences on the drawbacks of reanalysis data sets when the network of stations used for assimilation changes in space and time affecting the temporal and spatial co-variance patterns. p3. l. 1ff: In my opinion the authors could improve the intro by adding a short paragraph on their downscaling cascade from large to their localized scale. I guess that at least three levels of complexities are involved. An important issue in this context is that a consistent approach is desirable where the subsequent downscaling steps comprise over the at least as complex structure as the preceding. For instance, given the assimilated or GCM derived large-scale circulation shows too strong biases (e.g. blocking frequencies) also the following steps do not compensate this shortcoming but inherent the information from the boundaries. This should at least be kept in mind to consistently interpret results and according uncertainties.

[Figure]

2 Material and methods

2.2 Estimation of return level for regional maximum wind speeds

p4 l. 1ff: I assume that also a seasonal component is into the variability of maximum wind speeds. The authors could add some information which processes drive maximum wind speed during different seasons ( e.g. frontal based cyclonic maximum wind speed in cold fronts during winter half year vs. wind gusts originating from thunder storms that are operating during the summer seasons).

Another important information relates to the temporal basis. As much as I could infer authors use maximum monthly wind speeds. Using maximum daily wind data would provide a better statistical basis. However, in this case one also needs to account for the effect of serial correlated data.

A third issue involved in the analysis of extremes relates to the procedure of averaging – Are the values used for comparisons based on 6(x)hourly means or are they related to certain reading hours, i.e. instantaneous measurements without any temporal averaging ? This could for instance explain already part of the differences between ERA40 and observations. Given the comparable short length of the observational basis with the high value of return period it might also be useful to calculate shorter term return period, i.e. two and five years.

Discussion and conclusion:

How do other climate change studies (e.g. Bärring et al. 2017) addressing other climatic variables compare to changes that are potentially controlled by changes in extreme wind speeds?

Figure and Tables:

In general, on the small scale geographic information is missing at borders. It would also be helpful to include an inset covering the large scale surroundings. In addition, each map should have its own frame with lat/lon coordinates.

Table 1:

Please include the length of the individual meteorological recordings to better visualize the robustness in the estimation of the 10yr return period. If the length between the ERA40 and the meteorological station varies then only the common overlap period should be used. Another question is whether the direction of strongest wind direction is the same for both, the ERA40 data set and the meteorological observations, respectively.

Appendix Figure 1: For the comparison a similar basis should be used. Obviously the ERA40 data are based on maximum monthly wind speed whereas the boxplot is based on 10min readings. Again, it would be important to know the averaging procedure, especially for the Era40 interim data set.

Suggested Reference:

Bärring, L., Berlin, M. and B.A. Gull (2017) Tailored climate indices for climate-proofing operational forestry applications in Sweden and Finland, International Journal of Climatology, 37, 10.1002/joc.4691, 123–142.

Suggestion for modified abstract:

Abstract The bioeconomy has an increasing role to play in climate change mitigation and the sustainable development of national economies. In a forested country, such as Finland, over 50% of its current bioeconomy relies on the sustainable management and utilization of forest resources. In this paper, we examine the feasibility of the wind multiplier approach for downscaling of maximum wind speed, using 20 meter spatial resolution CORINE-land use dataset and high resolution digital elevation data. A coarse spatial resolution estimate of the 10-year return level of maximum wind speed was obtained from the ERA-Interim reanalysed data. These data were [Using a geospatial re-mapping technique the data ] were downscaled to 26 meteorological station locations to represent very diverse environments typical for Finish landscape.

Applying a comparison, the downscaled 10-year return levels represent 77% of the observed variation among the stations examined. In addition, the spatial variation of wind multiplier downscaled 10-year return level wind was compared with the WAsP-model simulated wind. The heterogeneous test area was situated in Northern Finland, and it was found that the major features of the spatial variation were similar, but in the details, there were relatively large differences. The results indicate that the wind multiplier method offers a pragmatic and computationally feasible tool for identifying at a high spatial resolution those locations having the highest forest wind damage risks. It can also be used to provide the necessary wind climate information for wind damage risk model calculations, thus making it possible to estimate the probability of predicted threshold wind speeds for wind damage and consequently the probability (and amount) of wind damage for certain forest stand configurations.

———————————————————

---

## Author Comment (AC1) · 4 May 2017

We would like to thank the reviewer for the constructive and positive comments that help us to improve the manuscript. The first comment is related to calculation of weighting factors "p. 4, l. 14: the rationale behind the weighting factor should be explained"

In our work we first interpreted the terrain cover available from CORINE database to surface roughness values using the same methods as used e.g. in the Wind Atlas (Tammelin et al. 2013). We are interested in very high resolution spatial variation of wind speed in typically highly variable terrain mosaic composed of forests, fields,

lakes, clear cut areas etc. The detailed structure of wind flow in this kind of heterogeneous terrain is very complex (e.g. Dupont&Brunet, Forestry, Vol. 81, No. 3, 2008. doi:10.1093/forestry/cpn006); one dominant feature being rapid deceleration of wind when wind encounters forest edge. The main wind damage are found typically within distance less than 50 m from the forest edge (Peltola et al. 1999b). In integration of the so called effective roughness we have applied normal distribution having variance 150 m. With these assumptions the weighting of each grid is as demonstrated in Fig. 1. The weight of the closest grid square is about 11 % and the furthest grid square located 500 m upwind has the weight of 0.04 % only. With no doubt, this formula is a simplification of a very complex issue as the exact impact of roughness elements on wind flow depend beside terrain properties also on the characteristics of prevailing air flow. However, when aiming in computationally light applications all these issues cannot be taken into account and the approach selected here gives a realistic interpretation of the complicated issue. We would be happy to add the Fig. 1. (e.g. to appendix) and explanatory text to the manuscript.

The next comment, "p. 6, l. 33-34: the authors use 12 m/s as a wind speed likely to cause damage. Given the availability of the HWind model, it would seem interesting to provide examples of stands that would be vulnerable to such wind speeds. The same applies at l. 39. I believe the paper could be strengthened if this section was somewhat expanded."

Simulations using HWind would really give additional value for the study. Unfortunately for this study and for the Pyhätunturi test-area we did not have all the needed forest data. However, in our next studies we plan to expand our studies to cover whole country is respect of wind climate and apply HWind model for another test areas where we have needed detailed forest characteristics data.

The third comment relates to accuracy of weather station data " p. 7, l. 28-29. The authors point out some potential imprecisions of weather station data. It would be interesting to know to what extent this problem is present in the data base."
FMI has a three stage quality control system, first check is done at the observation station site checking the main instrument malfunctions, and next check is done before storing the data to database. This check includes e.g. comparison with the extreme values and temporal and spatial consistency. The final step is the manual quality control for those values that did not pass earlier steps. The quality control ensures that the values stored in database are realistic and can have occurred. However, quality control does not guarantee that the measurements are exactly correct. As well, quality control does not ensure the homogeneity of observations. The changes at measuring site and changes in instrumentation as well as, the changes of the height of anemometer installation can lead to discontinuations, break points, in observational time-series. These break points are relatively common also in wind observational time series like studied by Laapas and Venäläinen (2017). We are happy to add some more text about the reliability of wind observations.

Next comments: "p. 13: please provide units with column headings". Yes, we will add. "p. 19: the title of the y axis should be changed since it represents 10 year return levels of maximum wind speed for two different approaches". Yes, we will change. p. 22: it is mentioned that the figure includes only values > 11.4 m/s. Do you mean that the graphs were truncated at this value or that the whole statistics did not consider lower values? The whole statistics did not consider lower values. This was done because we were interested in high wind speed values and the direction distribution of these strongest winds.

---

## Author Comment (AC2) · 4 May 2017

We would like to thank for the reviewer for the comments that help us to improve the manuscript.

The first comments is related to the Title "The title includes a bit more promises than actually addressed in the body of the manuscript – for risk management an assessment of potential future changes is necessary. Investigations about future scenarios are however not addressed in this study. Therefore I suggest to modify/correct the title for this missing part of the analysis."

[Figure]

The Title is "Estimation of the high-resolution variability of extreme wind speeds for a better management of wind damage risks to forest-based bioeconomy". If needed, we can shorten the title and make it more general e.g. "Estimation of the high spatial resolution variability of extreme wind speeds for forestry applications"

The next comment is related to Abstract. "The abstract is very extensive – At the end of the review the authors find a suggestion for a shortened version with focus on the very background and the most important findings."

We thank for the good text and are happy to edit the Abstract.

Introduction The comment: "p1. l. 5: could you please add a few examples from the papers you cite which specific risks are impacting on the forests in Finland."

The foreseen risks include increased wind throw risk due reduced soil frost period and depth. As well, drought may have negative impacts especially in southern Finland spruce forests. Related to drought, forest fire danger will increase. During winter season heavy snow loads will decrease in the southern but increase in the northern Finland. We will edit the text taking into account the comment.

Comment: "p2. l. 35ff: The authors should add one or two sentences on the drawbacks of reanalysis data sets when the network of stations used for assimilation changes in space and time affecting the temporal and spatial covariance patterns."

Yes, we will add some text about the factors affecting the accuracy of re-analyzed data.

Comment: "p3. l. 1ff: In my opinion the authors could improve the intro by adding a short paragraph on their downscaling cascade from large to their localized scale. I guess that at least three levels of complexities are involved. An important issue in this context is that a consistent approach is desirable where the subsequent downscaling steps comprise over the at least as complex structure as the preceding. For instance, given the assimilated or GCM derived large-scale circulation shows too strong biases (e.g. blocking frequencies) also the following steps do not compensate this shortcoming but inherent the information from the boundaries. This should at least be kept in mind to consistently interpret results and according uncertainties. "

This again is a relevant comment and we will add the paragraph. There are many challenges when downscaling to this high resolution. The coarse resolution re-analyzed data contain inaccuracies. This is concrete especially if we are looking specific weather events. Fortunately the mean and as well, return levels are not as much dependent on imperfectly simulated cases as if we were making cases-studies of e.g. on some disastrous event. However, if the model contain systematic biases then the downscaling we have applied is not able to correct them. The importance of small scale spatial variation is demonstrated in our study; the coarse scale ERA Interim based spatial variation of e.g. 10-year return level of maximum wind speed for the area covering the whole Finland is less than the simulated high resolution spatial variation within the small test area.

Material and methods The next comment related to Material and methods chapter, 2.2 Estimation of return level for regional maximum wind speeds "p4 l. 1ff: I assume that also a seasonal component is into the variability of maximum wind speeds. The authors could add some information which processes drive maximum wind speed during different seasons (e.g. frontal based cyclonic maximum wind speed in cold fronts during winter half year vs. wind gusts originating from thunder storms that are operating during the summer seasons)."

Yes, we will add text about the reasons for the occurrence of high wind speeds.

Comment: "Another important information relates to the temporal basis. As much as I could infer authors use maximum monthly wind speeds. Using maximum daily wind data would provide a better statistical basis. However, in this case one also needs to account for the effect of serial correlated data. A third issue involved in the analysis of extremes relates to the procedure of averaging – Are the values used for comparisons based on 6(x)hourly means or are they related to certain reading hours,

i.e. instantaneous measurements without any temporal averaging ? This could for instance explain already part of the differences between ERA40 and observations. Given the comparable short length of the observational basis with the high value of return period it might also be useful to calculate shorter term return period, i.e. two and five years."

In case of the re-analyzed coarse resolution data we have used 6-hourly data and the parameter was instantaneous 10-minute wind speed. When calculating the return levels we have applied the GEV approach that fits the extreme value distribution based on the annual maximum values. The same approach was used also for station data but now the parameter was the 3-hourly (synoptic) instantaneous 10-minute wind speed. In this sense the observational values are not exactly the same as reanalyzed data and this may create some systematic difference. However, when we finally use the annual maximum values as the bases for fitting the distribution this may reduce the bias. We can add discussion about this into the text. It is true that the estimation of e.g. 50-year return levels based on 30-years of data leads to large uncertainty in estimates. The shorter return periods like two and five years could be estimated more accurately but from the point of forestry planning activities the longer return periods are interesting and that is why we have demonstrated the applicability of the method with 10-year return level.

Discussion and conclusion

Comment: "How do other climate change studies (e.g. Bärring et al. 2017) addressing other climatic variables compare to changes that are potentially controlled by changes in extreme wind speeds?"

Bärring et al. (2017) examined climate change impacts on temperature and precipitation related indices relevant from the point of the Scots pine transfer functions. Their study is an interesting example on how climate change may influence on forestry. The importance of climate change impact studies is emphasized by the long rotation period

of 50-100 years from forest cultivation to final harvesting in Scandinavia. Bärring et al. (2017) found clear signal in temperature related indices but minor in precipitation and future climate is in that sense more favorable for forest growth. Bärring et al. (2017) did not study possible changes in wind climate that might influence on wind throw risks. There are other studies like Nikulin et al. (2011); Pryor et al, 2012; Outten&Esau, 2013) that indicate no clear signal in the occurrence of extreme winds in Scandinavia. We are happy to add discussion and references to Bärring et al. (2017) and other relevant literature into the manuscript.

References. Outten S.D. and Esau, I., 2013. Extreme winds over Europe in the EN-SEMBLES regional climate models. Atmos. Chem. Phys., 13, 5163–5172, 2013, www.atmos-chem-phys.net/13/5163/2013/, doi:10.5194/acp-13-5163-2013.

Nikulin G, KjellstroÂÍm E, Hansson U, Strandberg G, Ullerstig A., (2010) Evaluation and future projections of temperature, precipitation and wind extremes over Europe in an ensemble of regional climate simulations. Tellus Ser A Dyn Meteorol Oceanogr. doi:10.1111/j.1600-0870.2010.00466.x.

Pryor et al., 2012. Analyses of possible changes in intense and extreme wind speeds over northern Europe under climate change scenarios. Climate Dynamics, Vol. 38, No. 1-2, 2012, p. 189-208.

Figure and Tables: Comment: "In general, on the small scale geographic information is missing at borders. It would also be helpful to include an inset covering the large scale surroundings. In addition, each map should have its own frame with lat/lon coordinates."

There is information about the land use in Appendix, Figure A2. There exist detailed photos available e.g. in the Finnish National Surveys web service (https://asiointi.maanmittauslaitos.fi/karttapaikka/?lang=en) or Google Maps, we could add links to these services, however, we are hesitant to add these photos e.g. due to the copyright restrictions. We will add new information and the co-ordinates; this will

improve the reading of manuscript.

Comment: "Table 1: Please include the length of the individual meteorological recordings to better visualize the robustness in the estimation of the 10yr return period. If the length between the ERA40 and the meteorological station varies then only the common overlap period should be used. Another question is whether the direction of strongest wind direction is the same for both, the ERA40 data set and the meteorological observations, respectively."

We will include information about the length of observational data timeseries and also about the reliability of measurements.

Comment: "Appendix Figure 1: For the comparison a similar basis should be used. Obviously the ERA40 data are based on maximum monthly wind speed whereas the boxplot is based on 10min readings. Again, it would be important to know the averaging procedure, especially for the Era40 interim data set."

ERA data is 6-hourly data and we can add a box plot that is drawn using that data.